# Effect of Micro-Textures on the Surface Interaction of WC+Co Alloy Composite Coatings

**Xin Tong [1], Yu Zhang [1] and Xiaoyang Yu [2,\*]**

1   Key Laboratory of Advanced Manufacturing and Intelligent Technology, Ministry of Education, Harbin University of Science and Technology, Harbin 150080, China
2   School of Mechanical Power Engineering, Harbin University of Science and Technology, Harbin 150080, China
\*   Correspondence: yxy1236987456@163.com

**Abstract:** The surface properties of alloys can be improved by coating their surfaces and adding a micro-texture. The effect on the surface properties of alloy composite coatings of adding a textured surface has not been addressed in previous studies. This study added a micro-texture to the surface of a WC+Co alloy AlCrN/AlTiSiN composite coating. The influence of the micro-texture's geometric parameters on the surface properties of the composite coating and its lifespan were studied in detail. First, the surface hardness and phases of various micro-textured composite coatings were analyzed to explore the effect of different micro-texture parameters on the surface properties. Then, a friction and wear test was conducted to establish a model that can predict the lifespan of a micro-texture and the influence of different micro-texture parameters on the surface friction of the composite coating. After that, the wear pattern of the composite coating and the relative action of the micro-texture were analyzed on the basis of the visible wear morphology. The results show that using a laser to add a micro-texture to the surface of a composite coating creates a hardened layer that increases the coating's surface hardness. Analysis of the surface phases of the composite coating showed that there are three principal types of grain on the surface, namely WC, CrN and TiN, with WC having the largest grain size. The main kind of wear on the surface of the composite coating was found to be abrasive wear, which can be reduced by the addition of a micro-texture.

**Keywords:** composite coating; micro-texture; prediction model



## 1. Introduction

WC+Co alloy has a wide range of industrial applications because of its wear resistance, hardness, tolerance to high temperatures, low susceptibility to oxidation, etc. [1–3]. Adding a composite coating to the surface of WC+Co alloy further enhances its surface properties. AlCrN/AlTiSiN composite coating has excellent resistance to oxidation, high temperatures and wear. It was found that adding an AlCrN/AlTiSiN composite coating to the surface of WC+Co alloy is more effective than adding an AlTiSiN coating on its own. Studies have also shown that adding a micro-texture to the surface of the coating can help to further reduce the friction and wear, thereby prolonging the service life of the coating and, thus, the service life of the WC+Co alloy. If the micro-texture fails, the friction and wear of the surface coating increase, and its lifespan is reduced. It is, therefore, important to improve our understanding of the lifespan of micro-textured composite coatings.

Liu [4] et al. studied the synergistic effect of adding a micro-texture and solid lubricating coating on the anti-friction and anti-wear properties of aluminum alloy surfaces. This showed that the joint effect of a micro-texture and graphite coating improved the tribological properties of an aluminum alloy surface. The effects of different laser parameters on the depth and microstructure of a micro-texture were investigated by Xi et al. [5]. They found that texture depth is positively correlated with the average power and negatively correlated with the scanning speed. However, excessive power reduces the depth. Xu

et al. [6] examined the influence of a high-temperature environment on the wear resistance and wear mechanism of micro-textured samples. Here, the tribological properties of the micro-textured samples were better than those of just a titanium alloy matrix under normal and high-temperature wear environments, with the micro-textured samples having a lower friction coefficient and a smaller degree of wear. Sun et al. [7] looked at the effect of adding a $MoS_2$/Ti electrojet deposition flux of different heights to a coating's morphology. In this case, having a micro-texture helped to reduce the friction coefficient and degree of fluctuation by storing the coating material. Wang et al. [8] successfully fabricated NiCo coatings on Inconel 718 substrates using pulsed laser fusion technology. The characteristics and performance of the coating, including its microstructure, phase constitution, hardness, friction coefficient, wear volume, wear morphology and residual stress, were investigated using a variety of methods. The hardness of the coating was 427.82 HV, 21.1% higher than that of the Inconel 718 substrate. The coating also had a higher resistance to wear than the substrate. Qian et al. [9] prepared an Fe-based coating on the surface of DH36 steel by means of laser cladding. X-ray power diffraction (XRD), a scanning electron microscope (SEM) and a micro-hardness tester were used to analyze the phase composition, microstructure and micro-hardness of the coating. The corrosion resistance of the Fe-based coatings and substrates was analyzed by placing them in artificial seawater and examining their polarization curves. The results showed that the coating had mostly bonded to the substrate. The microstructure of the bonding zone chiefly consisted of plane crystals and columnar crystals growing upwards, while the middle and upper zones consisted of fine dendrites due to the combined effect of the solid solution, carbide dispersion and fine grain strengthening of the alloy elements. The average micro-hardness of the coating was 1026.11 HV, which was about 5.21 times higher than that of the matrix, and the corrosion resistance of the coating was significantly improved. Xiao et al. [10] studied the heavy-load elastohydrodynamic lubrication performance of a friction pair with a surface coating and the synergistic effect of adding a texture. On the basis of a generalized Reynolds equation, they established an elastohydrodynamic lubrication model of a micro-textured coating–substrate system. They used this to discuss the elastic modulus and the effect of a triangular texture's depth, width and density on a system's elastohydrodynamic response. The results showed that there is an optimal micro-texture depth, width and density that delivers the best load-carrying capacity for a coated gear. Selection of an appropriate coating and micro-texture design can improve the tribological properties of gears and serve to prevent the failure of membrane-based systems. Huang et al. [11] examined the effect of adding a mesh micro-texture on the friction and wear properties of 316 L stainless steel microwires. They used reciprocating small-stroke fretting friction and a wear tester to conduct friction and wear tests and measure the surface friction and wear depth under different loads. It was found that adding a mesh texture to the surface of stainless steel microwires significantly improved their friction and wear properties. The depth of the micro-texture was an important factor affecting the wear resistance, and the wear depth was positively correlated with the external load.

It has, therefore, been established that adding a micro-texture to the surface of an alloy coating can reduce its friction and wear characteristics and improve its operating performance. However, research regarding the effects of adding a surface micro-texture on the surface properties of a composite coating is still limited. This study, therefore, took the problem of how to best add a surface micro-texture to a composite coating as its starting point. The effects of and interaction between different micro-texture preparation parameters and geometric parameters on the surface hardness and phases of a WC+Co alloy composite coating were analyzed, and friction and wear experiments were conducted to investigate the friction behavior and identify the most effective composite coating in relation to the surface properties and lifespan of the micro-textured coating.

The research reported in this paper can be applied to the design of better cutters to improve the efficiency of metal processing, thereby helping to provide significant savings in industrial production costs.

## 2. Experimental Materials and Methods

### 2.1. Preparation of Micro-Textured Composite Coating

In comparison to just AlCrN or AlTiSiN coatings, AlCrN/AlTiSiN composite coatings offer a lower friction coefficient, greater hardness, improved oxidation resistance and better thermal stability. An AlCrN/AlTiSiN composite was therefore selected as the coating for a WC+CO alloy in this study. As the conductivity of the Cr element is poor, the AlTiSiN layer was applied first, then the AlCrN layer. The thickness of both layers was the same [12–14].

The specimens used for the tests were WC+Co alloy cylinders. A micro-texture was added to the surface of the composite coating. To prepare the specimens for the addition of the micro-texture, they were first ground with sandpaper, then wiped clean with alcohol. A fiber laser (Zhengtian Fiber Laser, Beijing, China) was used to engrave the micro-texture. Any surface burrs resulting from the mechanical process were removed, and the specimens were cleaned with an ultrasonic cleaner. The principles associated with PVD coating and micro-texture preparation are shown in Figure 1. The laser power was 35 W, its scanning speed was 1700 mm/s, the number of scans was 7, the micro-texture diameter was 60 μm and the micro-texture spacing was 130 μm. PVD coating technology involves using a low-voltage, high-current arc in vacuum conditions. An accelerated electric field evaporates a gas discharge and the target material and ionizes the evaporated material and gas. The products of the reaction are deposited on the workpiece [15,16]. The micro-texture preparation and geometric parameters featuring in the interactive experiment are shown in Table 1.

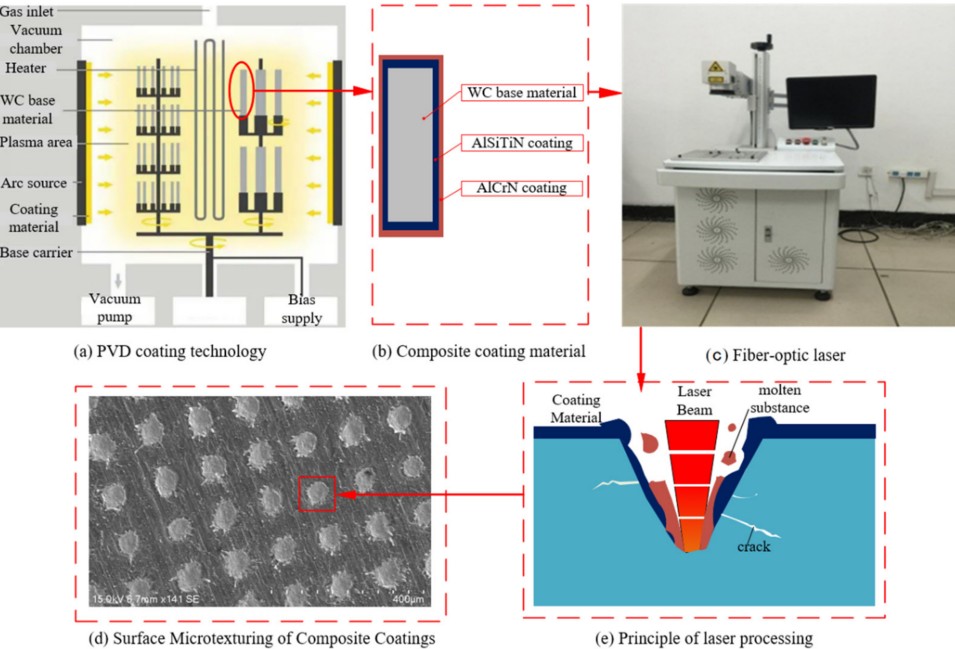

**Figure 1.** Coating technology and micro-texture preparation principles. (**a**) is PVD coating technology, (**b**) is Composite coating material, (**c**) is Fiber-optic laser, (**d**) is Surface Micro texturing of Composite Coatings, (**e**) is Principle of laser processing.

**Table 1.** Interactive test scheme.

| Factor<br>Level | Laser Power (W) | Scanning Speed (mm/s) | Scanning Times (Times) | Micro-Texture Spacing (μm) | Micro-Texture Diameter (μm) | $p \times d$ | $v \times d$ | $n \times d$ |
|---|---|---|---|---|---|---|---|---|
| 1 | 35 | 1500 | 6 | 130 | 40 | 1 | 1 | 1 |
| 2 | 40 | 1600 | 7 | 150 | 50 | 2 | 2 | 2 |
| 3 | 45 | 1700 | 8 | 170 | 60 | 3 | 3 | 3 |

### 2.2. Surface Friction and Wear Test

A friction and wear test was carried out on a WC+CO alloy sample with a micro-textured AlCrN/AlTiSiN composite coating using a CETR multi-functional friction and wear tester (Yicheng Hengda, Beijing, China), which was fixed to a chassis. Figure 2a,b shows the friction and wear test device. Two specimens were prepared and fitted one on top of the other into fixing discs, as shown in Figure 2d. During the test, data were captured using a vibration signal acquisition model, as shown in Figure 2c. The average load was set at 40 N, friction was applied for a period of 30 min, the maximum friction distance was 18 mm and the rotational speed was set at 100 m/min.

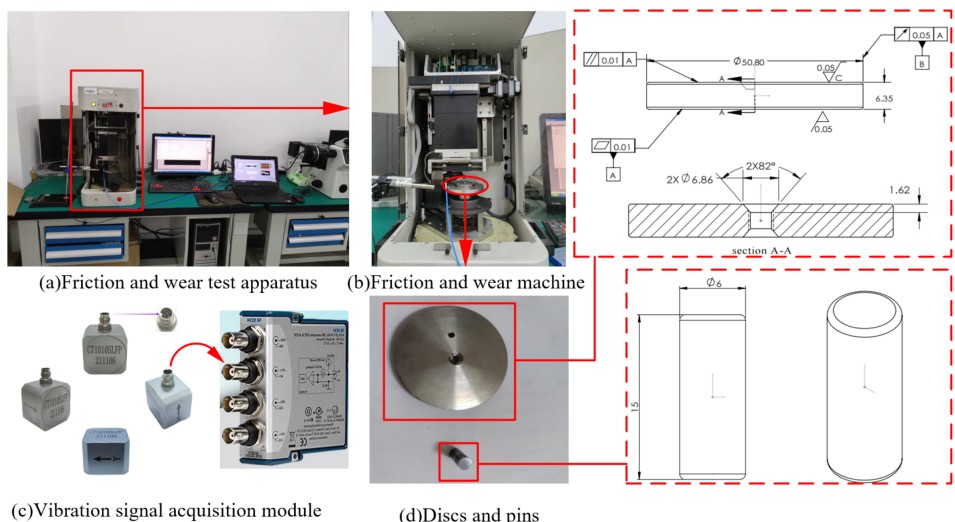

(a)Friction and wear test apparatus    (b)Friction and wear machine

(c)Vibration signal acquisition module    (d)Discs and pins

**Figure 2.** The friction and wear test device (**a–c**) and a schematic representation of the upper and lower samples (**d**).

### 3. Experimental Results and Analysis

*3.1. Surface Hardness of the Alloy Composite Coating*

The hardness of the machined surface was measured during each group of tests by a Digital Full Rockwell Hardness Tester (Heng Yi, Yantai, China). A test force of 1471 N was selected, which was applied by a diamond indenter. A variable-load handwheel on the device was turned clockwise to apply the total test force. The indenter was pushed into a spindle hole, close to the support surface. A notched plane in the indenter handle was placed against the setting screw, and the screw was tightened. Then, the HRC hardness block was positioned on the test bench. The handwheel was rotated, and the lifting screw rose so that the test piece was brought into contact with the indenter slowly, without any sudden impact. Once it was in contact, the test bench stopped rising, and the hardness tester automatically increased the test force, which was held for 5 s before being automatically removed. At this point, a buzzer sounded, and the hardness test value was displayed on the device's readout screen. A range analysis was performed on the hardness data measured during the tests, the results of which are shown in Table 2. All of the values given are Rockwell hardness values.

**Table 2.** Rockwell hardness values at different levels of micro-texture parameters.

| Rockwell Hardness | Laser Power (HRC) | Scanning Speed (HRC) | Scanning Times (HRC) | Micro-Texture Spacing (HRC) | Micro-Texture Diameter (HRC) | $p \times d$ (HRC) | $v \times d$ (HRC) | $n \times d$ (HRC) |
|---|---|---|---|---|---|---|---|---|
| First level | 77.111 | 77.706 | 77.328 | 77.417 | 77.128 | 77.3 | 77.611 | 77.42 |
| Second level | 77.45 | 77.194 | 77.267 | 77.689 | 77.489 | 77.51 | 77.289 | 77.46 |
| Third level | 77.8 | 77.572 | 77.767 | 77.256 | 77.744 | 77.55 | 77.461 | 77.47 |
| R | 0.689 | 0.512 | 0.5 | 0.433 | 0.616 | 0.25 | 0.322 | 0.05 |

From the results in Table 2, it can be seen that the interaction between the micro-texture preparation parameters and geometric parameters had little effect on the hardness of the surface coating. The main factors affecting the surface hardness of the coating were the laser power, $p$, and the micro-texture diameter, $d$.

When a laser acts on the surface of the coating, the high-energy beam makes the surface temperature of the composite coating rise rapidly until it reaches melting point, when local melting occurs. The heat transfer on the surface of the composite coating causes rapid cooling at the melting point, and a hardened layer forms on the surface. The rapid melting caused by the laser action generates pits, and the Cr element in the melted area and the C element in the cemented carbide chemically react to form a Cr–C compound. As the temperature drops, the Cr–C compound rapidly solidifies and spreads, thus, enhancing the surface coating phase. In other words, particle strengthening takes place that increases the hardness of the coating's surface.

After a micro-texture is placed on the surface of a composite coating, the surface area of the composite coating is reduced. When a load is then applied to the micro-textured area using a Rockwell hardness tester, the area is inversely proportional to the surface hardness value, so the hardness value becomes larger. As the micro-texture increases the mechanical properties of the composite coating's surface, the stress is more uniformly spread across it. When measuring the surface hardness of the composite coating, it is more difficult to indent than a comparable non-textured coating, so the textured surface of the composite coating improves its surface hardness.

### 3.2. Phase Analysis

A phase analysis of the composite coating surface was carried out using an X-ray diffractometer (XRD, Tong Da, Dandong, China). Phase data were collected and imported into JADE software (version 6.5) to analyze the phase composition. Figure 3 shows the phase structure of the surface of the textured composite coating. It can be seen that the X-ray diffraction pattern of the composite coating was dominated by peaks associated with the WC, TiN and AlN elements. No peak for the Cr element was detected. Due to the increase in the surface temperature of the composite coating during the preparation of the micro-texture, the diffusion rate of the Cr element was relatively fast, so any Cr elements at the surface of the coating were diffused into the interior of the composite coating or into the matrix, making the remaining Cr content at the surface relatively low. Any remaining Cr-containing grains were caught up within other grains after cooling. As a result, the surface contained no Cr phase data to be detected.

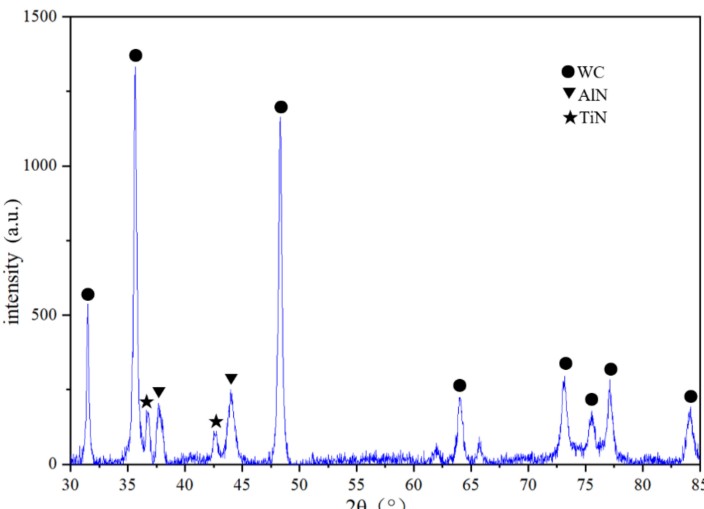

**Figure 3.** X-ray diffraction analysis of the composite coating.

The higher the peak intensity, as shown in Figure 3, the larger the grain size of the compound on the surface of the composite coating. If the grain size is too large, this increases the surface roughness of the composite coating, and its surface properties are degraded. A uniform and fine-grained structure is required to improve the surface properties of a composite coating. As the peak intensity fell at the same position for each element in each group of experiments, these peaks were subjected to range analysis.

The WC range analysis results are shown in Figure 4. It can be seen that the factors that had the greatest impact on the WC grain size were the laser power, *p*, and the scanning speed, *v*. Figure 5 shows the process that occurs when a laser acts on the surface of a composite coating. As already noted, the surface temperature of the composite coating increases rapidly. The WC inside the micro-texture is thermally decomposed into smaller grains. When the laser scanning speed increases, the depth of the prepared micro-texture becomes smaller, and there is less matrix material inside the micro-texture, so the size and number of WC grains decrease. If the laser power or the scanning speed is increased, it is beneficial for the refinement of the WC grains on the surface of the composite coating.

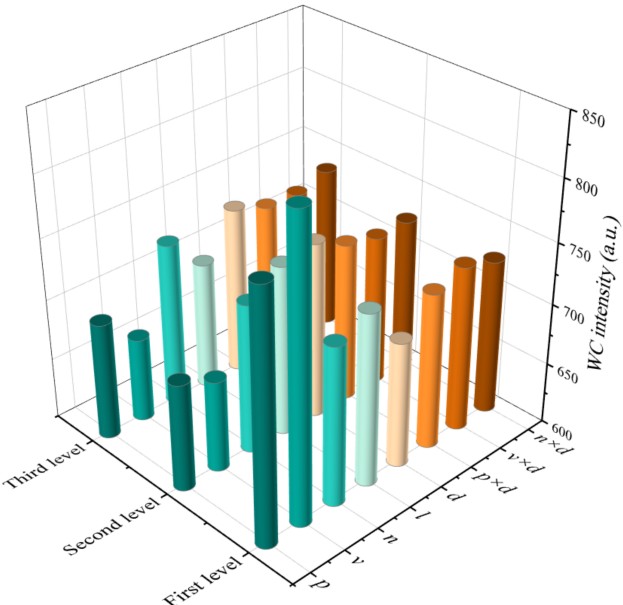

**Figure 4.** WC phase range analysis results.

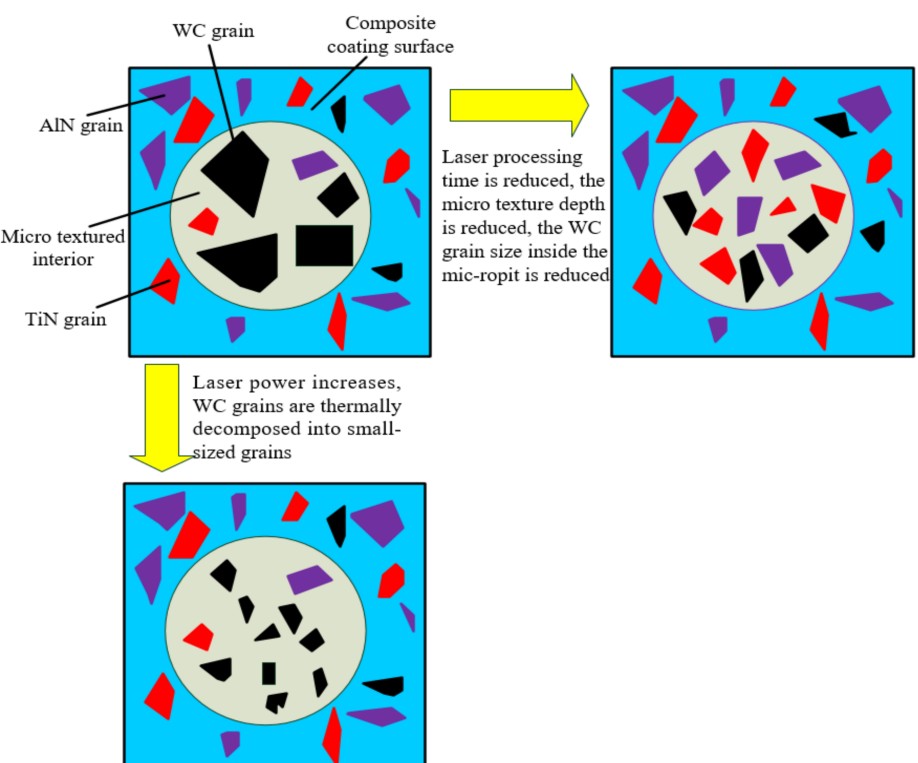

**Figure 5.** Changes in the WC grains resulting from laser action during the micro-texture preparation process.

When a laser acts on the surface of a composite coating, the surface temperature of the coating rises rapidly, the affected area melts and the Ti and Al elements in the composite coating react with the N in the molten pool to form TiN and AlN compounds. As the temperature drops, the TiN and AlN cool to form TiN and AlN grains.

It can be seen from the results of the TiN range analysis in Figure 6 that the factors that had the most significant impact on the TiN grain size were the scanning speed, $v$, and the laser power $p$. Compared with the WC grains, the size of the TiN grains changed less. When the laser scanning speed, $v$, increased, the time for the laser to act on the surface was reduced, making the WC grain size smaller so that some of the TiN grains took the place of the WC grains, increasing the size of TiN grains by the time the cooling finished. As the laser power increased, the N element on the surface of the coating was volatilized by the heat. This also affected the TiN grain size. So, an increase in laser power led to an increase in the WC content. Some of the TiN grains were also covered, further weakening the TiN peak.

From the results of the AlN range analysis shown in Figure 7, it can be seen that the factors that had the greatest influence upon the AlN grain size were the scanning speed, $v$, and the interaction between the laser power and the micro-texture diameter $p \times d$. This is because the diffusion rate of the Ti element on the surface of the composite coating was higher than that of Al element, so a place-exchange phenomenon occurred. Some Al elements were oxidized to form more uniform and dense tiny grains on the surface of the coating. This resulted in a reduction in the AlN grain size. As the laser power increased, the temperature in the molten pool increased, and the degree of mutual diffusion between each of the elements increased, resulting in a refinement of the final AlN grains. Some of the AlN grains were covered by other larger grains. It can be seen from Figure 8 that an increase in the micro-texture diameter and an increase in the laser power were together beneficial for the refinement of AlN grains on the surface of the composite coating.

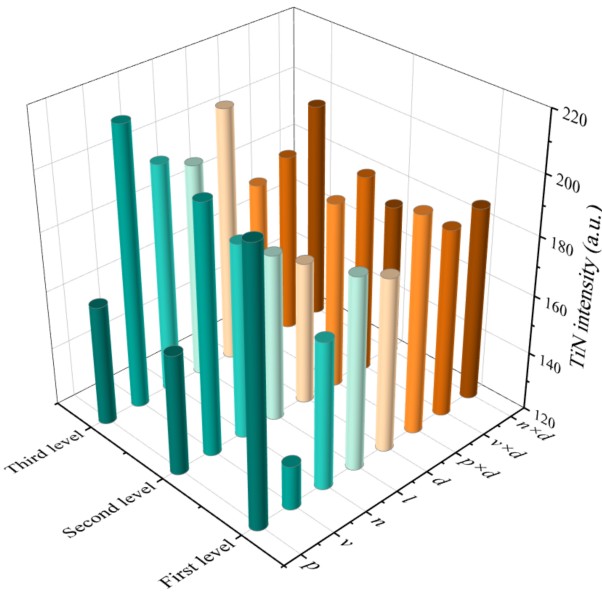

**Figure 6.** TiN phase range analysis results.

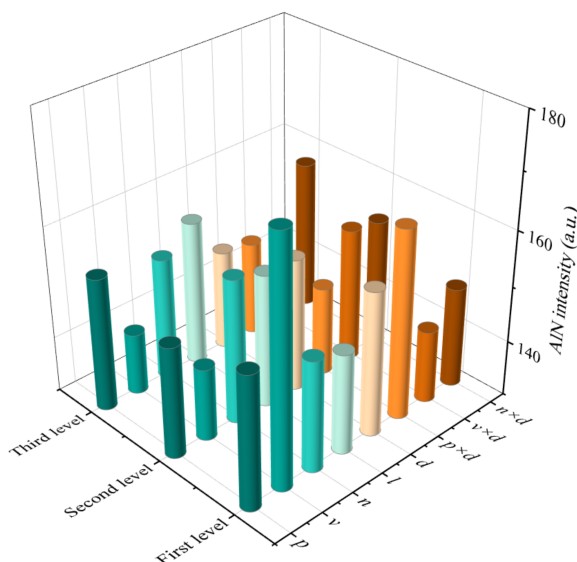

**Figure 7.** AlN phase range analysis results.

As the preparation and geometric parameters relating to the micro-texture changed, out of the three grains—WC, AlN and TiN—the WC grain size fluctuated the most, while the AlN and TiN grain sizes were relatively unaffected. For the purposes of grain refinement and improvement of the surface properties of a composite coating, it is, therefore, most important to augment the laser power and laser scanning speed within an appropriately determined range.

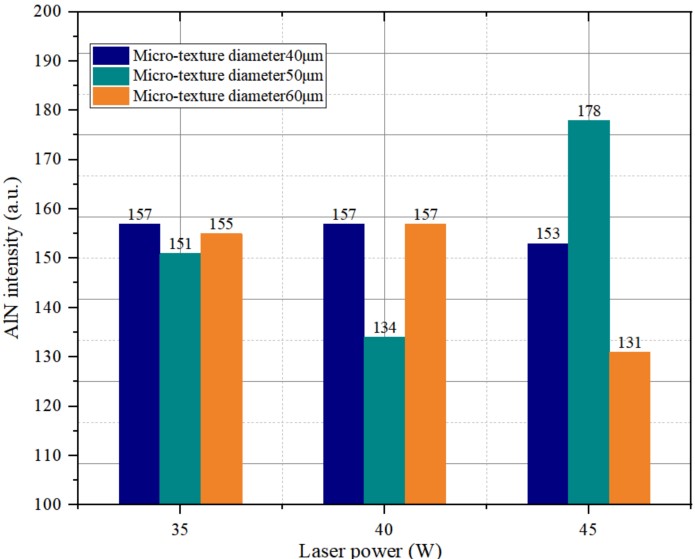

**Figure 8.** Binary interaction diagram.

### 3.3. Influence of Parameter Interactions on Friction Behavior of the Composite Coating Surface

Severe friction and wear in the area on the surface of a composite coating prepared with a micro-texture are the main reasons for a micro-texture to fail. Once the micro-texture is lost, this reduces the performance of the surface of the composite coating. The friction-reducing properties of micro-textures have a crucial impact on the surface properties of composite coatings, so the effects of different micro-texture preparation parameters and geometric parameters on their friction-reducing properties were analyzed.

Each of the samples was subjected to friction during the tests for a 30 min period. Figure 9 shows the variation in the values of the frictional force over time for each group of tests.

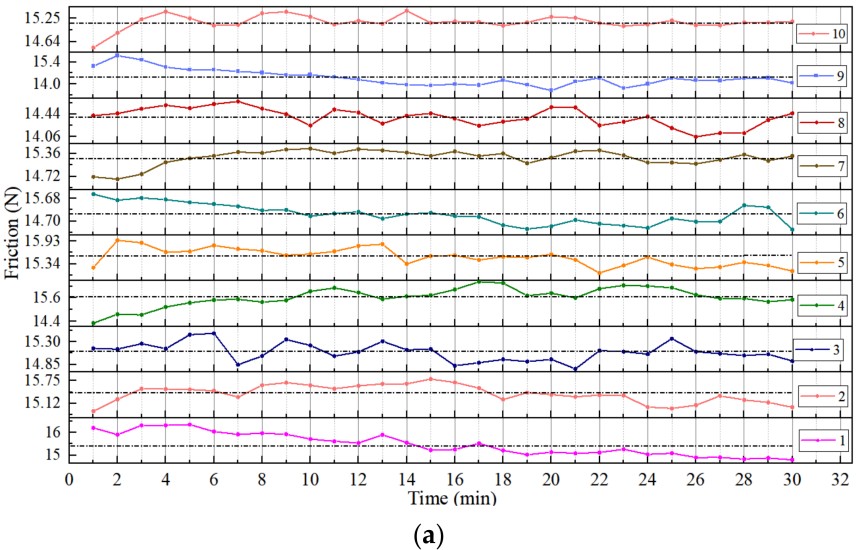

(**a**)

**Figure 9.** *Cont.*

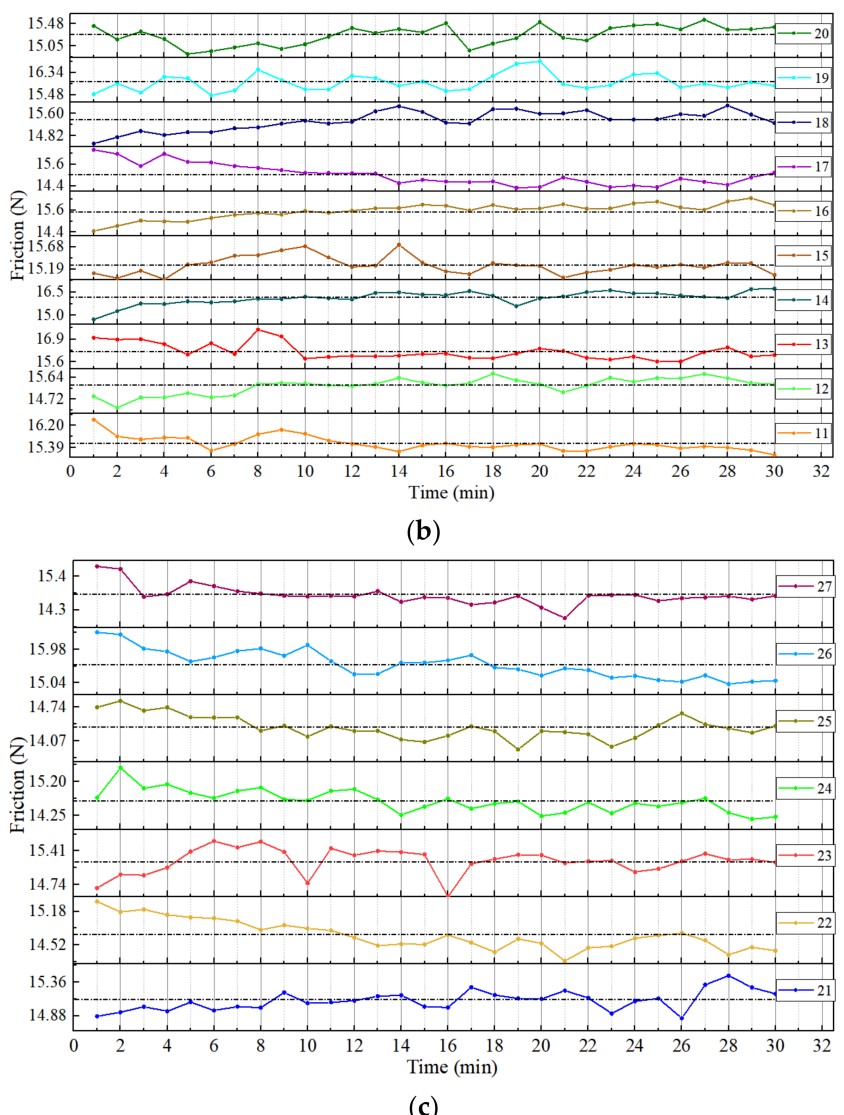

**Figure 9.** Frictional force values over the period 0–30 min. (**a**) is friction data for 1–10 sets of tests, (**b**) is friction data for 11–20 sets of tests, (**c**) is friction data for 21–27 sets of tests.

It can be seen from Figure 9 that the frictional force was relatively large when the friction and wear first occurred. The cracking of the coating on the test piece produced a number of abrasive particles. The adhesion of these particles on the surface of the test piece led to an increase in the friction coefficient. As the geometrical parameters of the micro-texture differed from those relating to the preparation, the running-in stage was also variable. As the running-in stage progressed, the friction began to decrease until a more stable wear stage [17,18] had been achieved. It can be seen from Figure 9 that the frictional force decreased over a period varying from 2 to 6 min, at which point the running-in stage was over, and the micro-texture began to play a role.

During the stable wear stage, the surface roughness of the specimen was relatively slight because some of the abrasive particles had been ground into fine powder. This fine powder became trapped inside the micro-texture, further reducing the friction coefficient of the surface. Some of the particles oxidized on the surface of the composite coating to form fine oxide particles. As the friction and wear test progressed, some of the coating began to peel off due to the increase in temperature. At this point, the surface became rougher again, and the frictional force increased.

Failure of the micro-texture due to long-term wear resulted in an increase in the frictional force. At this point, the workpiece entered into a stage of severe wear. Different micro-texture parameters resulted in the severe wear stage being reached at different times within each group of tests. It can be seen in Figure 9 that the micro-texture typically failed after between 18 and 22 min.

In Figure 10, the innermost ring shows the lowest surface friction value for each parameter at the first level. The outermost ring shows the lowest surface friction value for each parameter at the third level. The factors that had the most significant influence on the surface friction were the laser power, *p*, the micro-texture spacing, *l*, and the interaction between the number of laser scans and the micro-texture diameter, $n \times d$. This implies that these three factors are the most important ones to consider when seeking to control the surface friction of a composite coating. The laser acts on the surface of the composite coating. The material in the laser-treated area heats up and melts, forming splashes that cool on the surface of the coating. When the laser power increases, the time the laser beam acts on the surface of the coating is longer, the energy operating on the surface increases and the amount melted also increases. As a result, after cooling, the number of splashes on the surface is increased, the surface roughness is increased and the temperature is also higher during the process. This softens the surface of the coating, making it easier for the surface material to fall off, further adding to the surface roughness and the average frictional force. As the laser power increases, the depth of the micro-texture increases. When the surface is subjected to friction and wear, the ability of the micro-texture to capture and store wear debris is therefore enhanced. The size of the WC grains on the surface of the composite coating is significantly reduced, and the surface roughness is reduced. The surface friction of the composite coating, therefore, shows a downward trend as the laser power is increased. During friction and wear, the Al element also oxidizes on the surface of the coating to form an $Al_2O_3$ film. This has a lubricating effect and reduces the surface roughness. The laser power has an impact on this phenomenon by affecting the Al content of the coating's surface. The laser processing of the micro-texture produces burrs around the edges of the micro-pits, increasing the roughness in the areas where the micro-texture is present. When the micro-texture spacing is increased, the overall amount of micro-texturing on the surface decreases. There are, therefore, fewer burrs and less friction. This suggests that the micro-texture distribution should not be too dense and that an appropriate increase in the micro-texture spacing will help to reduce the surface friction.

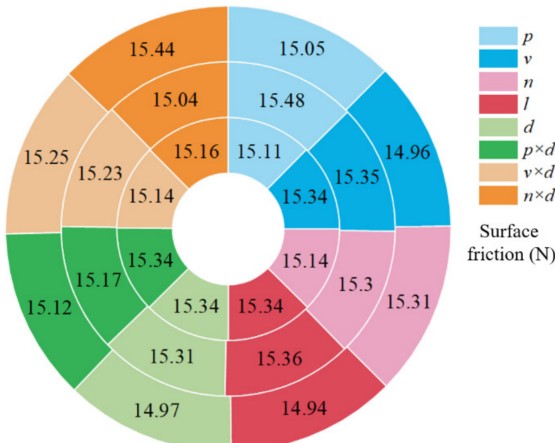

**Figure 10.** Range analysis of the average frictional force on the surface of the coating.

Figure 11 is a binary diagram relating to the interaction between the number of laser scans and the micro-texture diameter. It can be seen that the pattern of changes between the three groups was inconsistent. When the micro-texture diameter increased, the ability of the micro-texture to capture and store wear debris was enhanced, and the heat dissipation performance was better. The surface of the composite coating was not easily damaged, so

increasing the micro-texture diameter helped to enhance its anti-wear performance. As the number of laser scans increased, the laser energy acting on the surface increased. The number of spatters on the surface of the composite coating therefore also increased. The spatters cooled rapidly and adhered to the surface of the coating, resulting in a decrease in surface roughness. As the number of laser scans increased, the depth of the micro-texture also increased. This reduced how much material could splash out of the micro-texture. This enhanced the replenishment of the micro-texture chips. The Al oxidation on the surface also increased as the temperature rose, improving the lubrication effect and reducing the grain size and surface roughness. So, micro-texturing the surface of a composite coating dramatically reduces the surface friction. The frictional force decreased as the number of laser scans increased, and the diameter of the micro-texture became smaller. An increased depth leads to a decrease in a micro-texture's lifespan and creates an upward trend. Taking the surface hardness, element content, phase analysis and friction and wear test results together, it can be seen that the enhanced surface hardness of a composite coating and refinement of the surface grains induced by adding a micro-texture are all conducive to reducing surface friction.

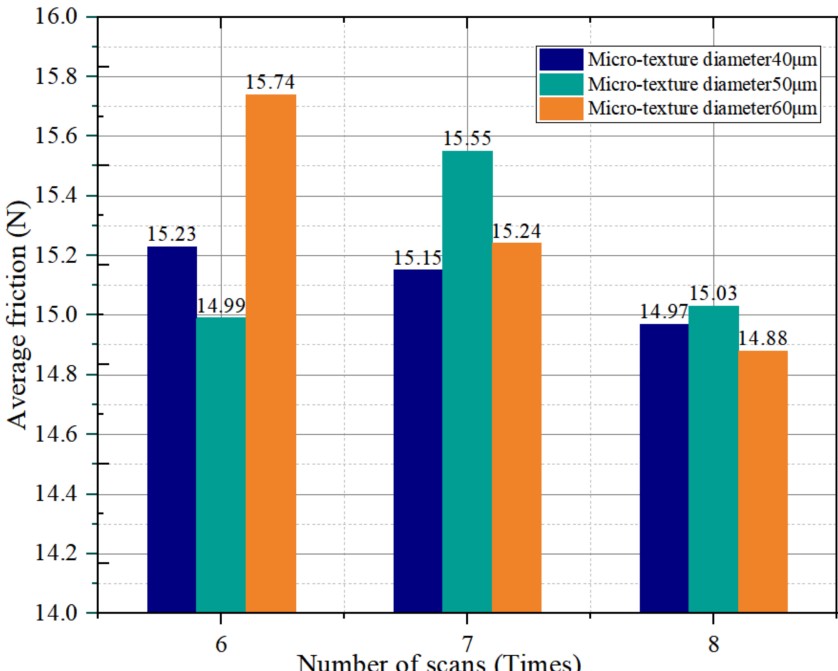

**Figure 11.** Binary diagram for the interactions.

Taking into account the surface hardness, phase and friction force of the composite coating, Table 3 shows the influence of the micro-texture preparation parameters on the surface properties of the textured composite coating. An analysis of the frictional force for each period of the friction and wear test in relation to the lifespan of the micro-texture is shown in Table 4.

**Table 3.** Influence of micro-texture parameters on surface properties.

| Laser power (W) | 35 | 40 | 45 |
|---|---|---|---|
| Coating surface properties | Worse | Average | Better |
| Laser scanning speed (mm/s) | 1500 | 1600 | 1700 |
| Coating surface properties | Worse | Average | Better |
| Number of laser scans (Times) | 6 | 7 | 8 |
| Coating surface properties | Worse | Average | Better |
| Micro-texture diameter (μm) | 40 | 50 | 60 |
| Coating surface properties | Worse | Average | Better |
| Micro-texture spacing (μm) | 130 | 150 | 170 |
| Coating surface properties | Worse | Average | Better |

**Table 4.** Micro-texture life cycle.

| Serial Number | Life Cycle (min) | Serial Number | Life Cycle (min) | Serial Number | Life Cycle (min) |
|---|---|---|---|---|---|
| 1 | 18 | 10 | 16 | 19 | 15 |
| 2 | 16 | 11 | 15 | 20 | 16 |
| 3 | 15 | 12 | 16 | 21 | 16 |
| 4 | 15 | 13 | 15 | 22 | 17 |
| 5 | 15 | 14 | 15 | 23 | 17 |
| 6 | 14 | 15 | 15 | 24 | 16 |
| 7 | 15 | 16 | 16 | 25 | 17 |
| 8 | 15 | 17 | 17 | 26 | 16 |
| 9 | 15 | 16 | 15 | 27 | 16 |

*3.4. Micro-Texture Lifespan Prediction Model*

On the basis of the friction and wear test, a micro-texture lifespan prediction model for composite coatings was established, with the micro-texture preparation and geometric parameters being used as independent variables. The mathematical model is:

$$T = Kp^{\alpha_1}v^{\alpha_2}n^{\alpha_3}l^{\alpha_4}d^{\alpha_5} \tag{1}$$

where $T$ is the lifespan of the micro-texture; $K$ is the coefficient relating to the micro-texture lifespan; $\alpha_1$, $\alpha_2$, $\alpha_3$, $\alpha_4$ and $\alpha_5$ are undetermined coefficients relating to the micro-texture parameters; $p$ is the laser power; $v$ is the laser processing speed; $n$ is the number of laser scans; $l$ is the micro-texture spacing; and $d$ is the micro-texture diameter.

Taking the logarithm of each side of Equation (1):

$$\lg T = \lg K + \alpha_1\lg p + \alpha_2\lg v + \alpha_3\lg n + \alpha_4\lg l + \alpha_5\lg d \tag{2}$$

Let $y = \lg T$, $c = \lg K$, $x_1 = \lg p$, $x_2 = \lg v$, $x_3 = \lg n$, $x_4 = \lg l$ and $x_5 = \lg d$ to obtain:

$$y = c + \alpha_{1\times1} + \alpha_{2\times2} + \alpha_{3\times3} + \alpha_{4\times4} + \alpha_{5\times5} \tag{3}$$

If we substitute data into Equation (3) to obtain a system of equations:

$$Y = XC \tag{4}$$

such that:

$$Y = \begin{bmatrix} y_1 \\ y_2 \\ \cdots \\ y_{27} \end{bmatrix}, X = \begin{bmatrix} 1 & x_{1-1} & x_{1-2} & x_{1-3} & x_{1-4} & x_{1-5} \\ 1 & x_{2-1} & x_{2-2} & x_{2-3} & x_{2-4} & x_{2-5} \\ \cdots & \cdots & \cdots & \cdots & \cdots & \cdots \\ 1 & x_{27-1} & x_{27-2} & x_{27-3} & x_{27-4} & x_{27-5} \end{bmatrix}, C = \begin{bmatrix} c \\ \alpha_1 \\ \alpha_2 \\ \alpha_3 \\ \alpha_4 \\ \alpha_5 \end{bmatrix}$$

Through matrix operations, we obtain:

$$C = (X^T X)^{-1} X^T Y$$

This gives a prediction model for the lifespan of the micro-texture that is based on the friction and wear test reported above:

$$T = 10^{1.2405} p^{0.1583} v^{-0.0153} n^{-0.2303} l^{-0.0207} d^{0.0052}$$

To ensure its reliability, actual micro-texture parameters were substituted into the model, and a significance test was carried out. The resulting variance analysis is shown in Table 5.

**Table 5.** ANOVA table of micro texture life cycle.

| Source of Variance | Degrees of Freedom | Sum of Square | Mean Square | F Value |
|---|---|---|---|---|
| Regression analysis | 5 | 10.0298 | 2.00296 | 6.710083 |
| Residual value | 21 | 6.2692 | 0.2985 | - |
| Total value | 26 | 16.299 | - | - |

If we raised the significance level to 0.05 to improve the accuracy, we found that $F_{0.05}(5,21) = 2.68$. As the actual value of F was 6.710083, which is much larger than 2.68, the reliability of this prediction model was confirmed to be significant.

*3.5. Surface Wear Analysis*

It can be seen from Figure 12 that friction and wear caused serious damage to the surface of the composite coating. The primary mode of the wear was abrasive wear. At the beginning of the test, the two friction surfaces were directly meshed. As the surfaces slipped relative to each other, scratches and abrasive particles were generated in the direction of the friction. Much of the surface of the composite coating adhered to a large proportion of the material that was peeling off and the oxides, so the primary mode of wear was, initially, viscous wear. As the wear increased, however, the number of abrasive particles on the surface of the coating began to increase. The increase in the surface temperature of the composite coating led to further oxidation of the surface material. The black particles in Figure 12 are oxides of the surface material and C oxides. Therefore, in addition to adhesive wear and abrasive wear, oxidative wear also features in the friction and wear process.

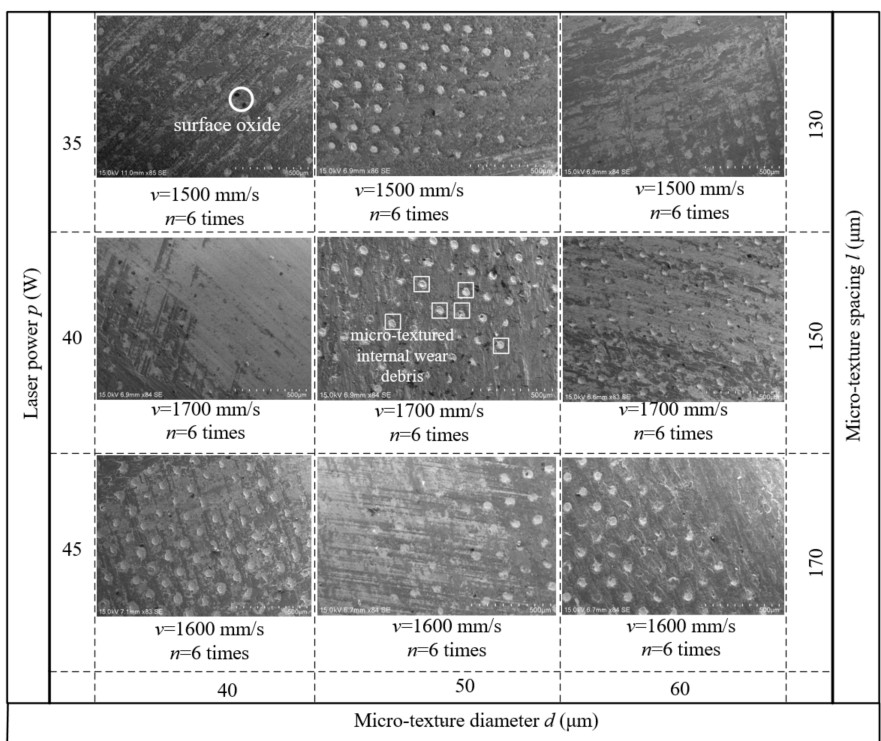

(a)Groups 1—9

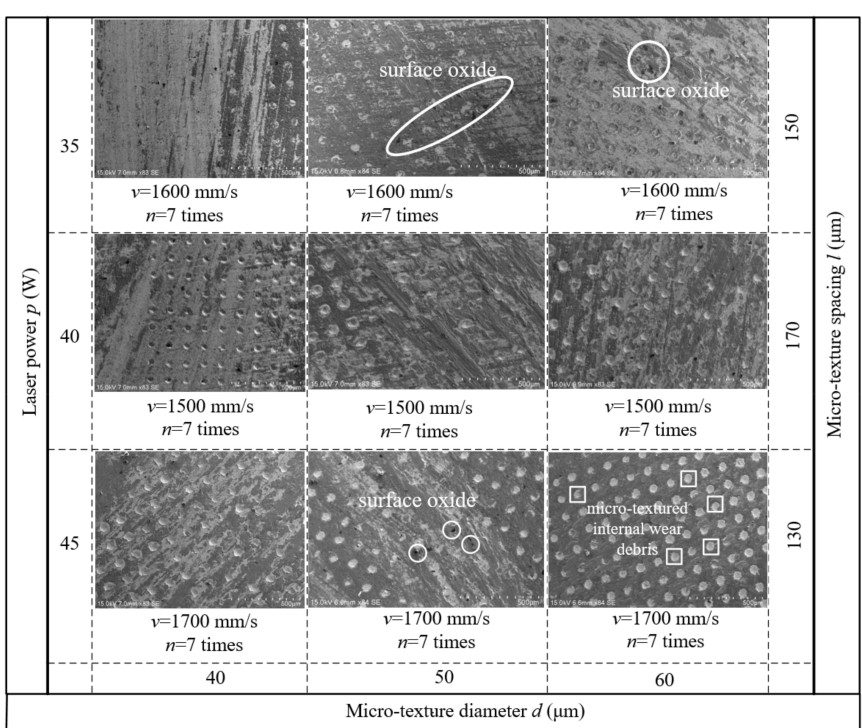

(b)Groups 10—18

**Figure 12.** *Cont.*

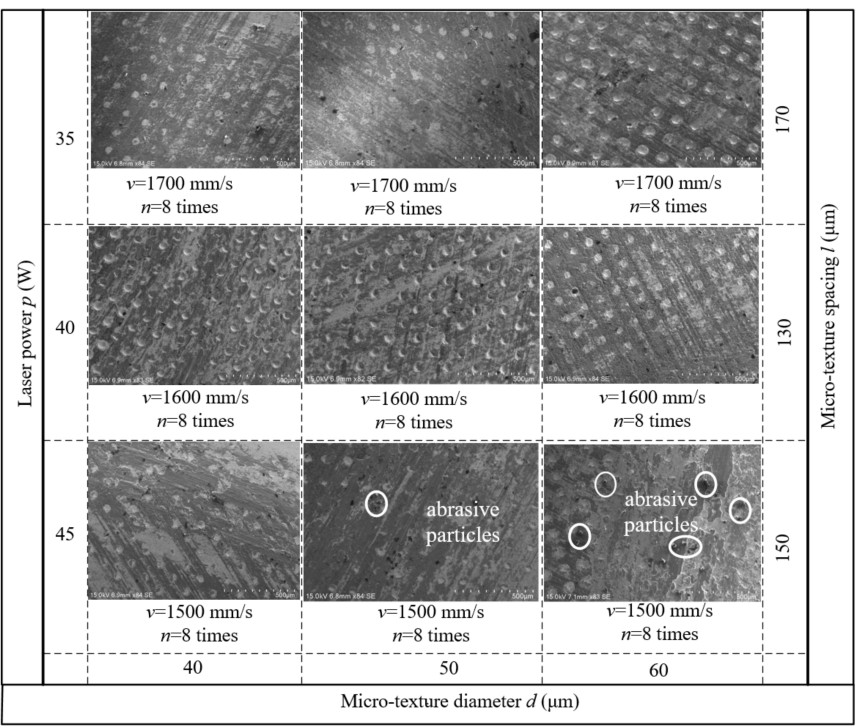

(c)Groups 19—27

**Figure 12.** Micro-texture wear morphology. (**a**) is the friction and wear morphology of 1–9 groups of tests, (**b**) is the friction and wear morphology of 10–18 groups of tests, (**c**) is the friction and wear morphology of 19–27 groups of tests.

The micro-texture was placed on the surface of the composite coating. This reduced the surface roughness of the coating and limited the number of adhesive nodes created during the friction and wear process, thereby reducing the amount of adhesive wear. It can be seen from Figure 13 that the micro-texture worked by storing the abrasive particles. Although the central area was seriously worn, most of the abrasive particles remained in the micro-texture, which reduced the wear debris on the surface of the coating and, thus, the abrasive surface wear. The placement of the micro-texture also increased the capacity of the surface of the composite coating to dissipate heat and reduced the oxidative surface wear. In sum, adding a micro-texture to the surface of a composite coating effectively reduces the degree of surface wear.

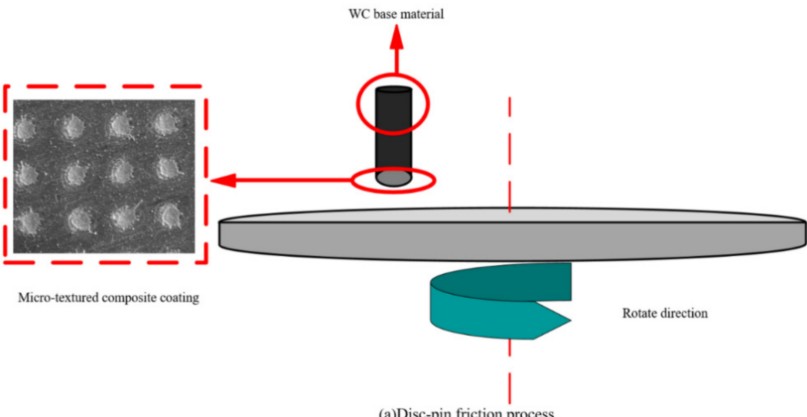

**Figure 13.** *Cont.*

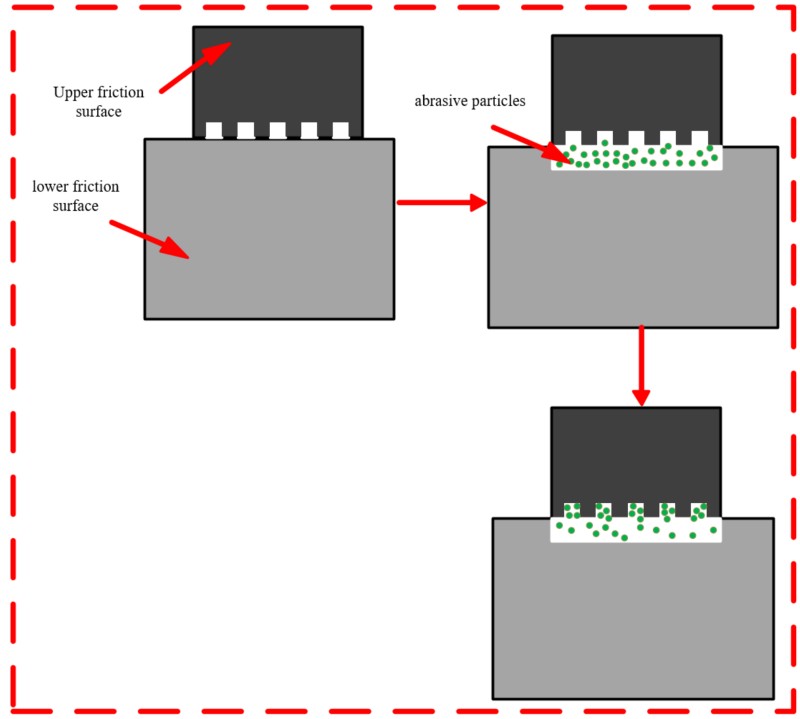

(b)The mechanism of action of micro-texture

**Figure 13.** Test process and micro-texture mechanism.

### 4. Conclusions

(1) When the laser power acting on the surface of a composite coating increases, the energy acting on the surface increases. A hardened layer is formed in the machined area to increase the surface hardness. The increase of the micro-texture diameter enhances the mechanical properties of the composite coating surface, resulting in enhanced surface hardness. There are three principal types of grain on the surface of composite coatings, WC, CrN and TiN. WC grains are much larger than the other two. When the laser power increases, the WC grains on the coating surface are thermally decomposed. The WC grains are refined. When the scanning speed increases, the grain size of TiN and AlN on the coating surface increases to cover some of the WC grains, resulting in a decrease in the measured grain size of the WC;

(2) An increase in laser power and micro-texture spacing helps to reduce the surface friction of micro-textured composite coatings. The interaction between the number of laser scans and the diameter of the micro-texture shows that increasing the number of scans serves to reduce the surface friction. Experimental analysis of the surface hardness, phase grain size and surface friction of the textured composite coating showed that the optimal processing parameters are: $p$ = 35 W; v = 1700 mm/s; $n$ = 8; $l$ = 170 μm; and $d$ = 60 μm;

(3) Further analysis of experimental friction and wear test results made it possible to construct a predictive model based on micro-texture lifespan. Observations of the micro-texture wear morphology revealed that the principal forms of wear on a composite coating's surface are viscous wear, abrasive wear and oxidative wear. Overall, adding a micro-texture to the surface of composite coatings helps to improve their resilience to surface wear.

**Author Contributions:** Conceptualization, X.T. and Y.Z.; methodology, X.T.; software, X.Y.; validation, X.T., Y.Z. and X.Y.; formal analysis, Y.Z.; investigation, X.T.; resources, X.T.; writing—original draft preparation, Y.Z.; writing—review and editing, Y.Z.; funding acquisition, X.T. All authors have read and agreed to the published version of the manuscript.

**Funding:** This work was funded by the Youth Science Foundation Project of China (Study on the Milling Behavior of Titanium Alloys by Ball End Mills and the Surface Integrity of Workpieces under the Synergistic Action of Mesoscopic Geometric Features and Coatings (52005140)), China.

**Institutional Review Board Statement:** Not applicable.

**Informed Consent Statement:** Not applicable.

**Data Availability Statement:** Not applicable.

**Acknowledgments:** The authors give thanks for the funding from the China Youth Science Foundation Project and express gratitude to EditSprings (https://www.editsprings.cn, accessed on 20 July 2022) for the expert linguistic services provided.

**Conflicts of Interest:** The authors declare no conflict of interest.

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
