# Peer review of "Effect of Micro-Textures on the Surface Interaction of WC+Co Alloy Composite Coatings"

_coatings, doi:10.3390/coatings12091242_

Round 1

Reviewer 1 Report

Dear Authors,

Congratulations on your work, which is focused on a very interesting subject. As any other paper in this phase, there are some amendments to do, whose can improve the overall quality of your paper. Thus, I'm providing below some comments and suggestions, trying to collaborate by this way in improving your paper:

1. The Abstract doesn't clearly state the literature gap found, as well as the main motivation to develop this work. Thus, please clearly state the gap found in the literature in the Abstract, Introduction and Conclusions. The mains goals are also not clear in the Abstract.

2. The novelty brought by your work is also not properly pointed out. Thus, please state clearly the novelty that your paper represents for the scientific community, stating as well if your contribution is exclusively scientific or if there was some practical motivation behind the development of your work. Any industrial application based on this work should also be pointed out.

3. The Literature Review is well done, but readers prefer direct speech, describing briefly in what the work of previous Researchers has been focusedon, methodology used and main results as you are doing with reference [5], [6] and [7]. Please avoid as much as possible generic ideas.

4. The number of references and the background regarding previous works is poor, thus, it must be improved. I'm suggesting the inclusion of some often cited papers in the field of coatings and machining, such as: doi: 10.3390/coatings8110402; doi: 10.3390/coatings10030235; doi: 10.3390/met10020170; doi: 10.3390/coatings6040051; doi: 10.1007/s00170-016-9514-3; doi: 10.1016/j.acme.2014.05.001; doi: 10.1007/s00170-019-03351-8; doi: 10.3390/met11020260; doi: 10.1016/j.jclepro.2020.121160; doi: 10.1016/j.triboint.2014.09.002; doi: 10.1016/j.wear.2021.203695; 10.1016/j.jmapro.2019.05.021; doi: 10.1016/j.ceramint.2016.10.151; doi: 10.1016/j.ceramint.2018.11.055.

5. Maybe section 2 can be converted in MATERIALS AND METHODS section, because this section is missing and some information usually contained in that section is dispersed by section 2.

6. Some results are dispersed between section 2 and section 3. Please reorganize the information in a traditional way, such as: Abstract, Introduction, Materials and Methods, Results, Discussion, Conclusions, References.

7. Please align the results in Table 2.

8. Figure 15 and Figure 16 are too small to allow a perfect observation of the surfaces' texture. Please enlarge, improving the quality.

9. No discussion is provided around the Results. Moreover, Results are not focused on the essential. Thus, please include discussion, previously properly organizing the results.

10. English should be revised by a NATIVE english Speaker to improve the readability.

Best wishes.

Kind regards

Reviewer 2 Report

I do not recommend the article for publication in its current state.

The topic can be the direction of important and truly key developments. Therefore, it would be good to give this article a new look.

The English of the article is not good, understanding is very difficult, sometimes almost impossible for me. Also there are wrong labels on the graphs as well.

The interpretation of the results is also not appropriate, because it does not show the reader what it wants, simply show the data.

Reviewer 3 Report

I miss the manufacturers, descriptions and types of the used analyses instruments and equipements . I find the references to be few. Comments and Suggestions for Authors were added in the attachment.

Round 2

Reviewer 1 Report

Dear Authors,

Congratulations for the improvements made on your paper, whose are in line with my expectations. Thank you so much.

Kind regards.

Author Response

Dear Reviewer
       Thank you very much for your valuable comments and for your recognition of my manuscript.

Reviewer 2 Report

Thank you for the improvement, now the paper is much consumable. This is important, in my opinion, since authors want other researchers to understand their findings. 

Please, read the article carefully one more time, without hurry, check the tables, and the sentences.

For example

In table 1 the laser power is counted in watts. The right abbreviation is W instead of w. 

I miss the units in table 2: laser power, scanning times, spacings... 

I do not feel the importance of figure 3. If you really want to keep this in the article, please replace the pictures by new ones. 

Figure 8 z axis, TiN intensity (SPACE) (counts)

Figure 9 again z axis

Figure 10 y axis tha same

Figure 11 x and y axis the same, please increase the label font size for better visibility

Figure 13: insert a space between x and y axis label name and unit 

Figure 14:

-spaces between text and unit

- laser scanning power? There is scanning speed and laser power. I feel disturbed when reading laser scanning power. 

In the conclusions: 

1) You increased the power, the scanning speed and the repetition time simultaneously. In my opinion this is not the right way to find some real effect, but anyway, this investigation can used as well, if you calculate the energy, which was absorbed to the material. When you increase the power and also the scanning speed, the energy can be the same. If you want to draw real conclusions, this is necessary. I cannot accept your conclusion 1 in its present form. 

2) it would be better to see a graph showing that p=35 W, v=1700 mm/s, n=8, l=170 um and d=60 um is the best value. I suggest to calculate the absorbed energy for all of your settings and show it in the function of friction force (or similar). 

Energy is always a key, since laser treatment make some changes in your surface but also beyond the surface because of heat. A predictive model should take both of this two into account. 
